# “In a Way We Took the Hospital Home”—A Descriptive Mixed-Methods Study of Parents’ Usage and Experiences of eHealth for Self-Management after Hospital Discharge Due to Pediatric Surgery or Preterm Birth

**DOI:** 10.3390/ijerph18126480

**Published:** 2021-06-15

**Authors:** Rose-Marie Lindkvist, Annica Sjöström-Strand, Kajsa Landgren, Björn A. Johnsson, Pernilla Stenström, Inger Kristensson Hallström

**Affiliations:** 1Department of Health Sciences, Faculty of Medicine, Lund University, 221 00 Lund, Sweden; annica.sjostrom-strand@med.lu.se (A.S.-S.); kajsa.landgren@med.lu.se (K.L.); inger.kristensson_hallstrom@med.lu.se (I.K.H.); 2Department of Computer Science, Lund University, 221 00 Lund, Sweden; bjorn_a.johnsson@cs.lth.se; 3Department of Clinical Sciences, Pediatrics, Lund University, 221 00 Lund, Sweden; pernilla.stenstrom@med.lu.se; 4Department of Pediatric Surgery, Skåne University Hospital, Lasarettsgatan 48, 221 85 Lund, Sweden

**Keywords:** app, eHealth, home-based care, hospital-to-home transition, neonatal care, pediatric surgery, preterm born, tablet

## Abstract

The costly and complex needs for children with long-term illness are challenging. Safe eHealth communication is warranted to facilitate health improvement and care services. This mixed-methods study aimed to describe parents’ usage and experiences of communicating with professionals during hospital-to-home-transition after their child’s preterm birth or surgery for colorectal malformations, using an eHealth device, specifically designed for communication and support via nurses at the hospital. The eHealth devices included the possibility for daily reports, video calls, text messaging, and sending images. Interviews with 25 parents were analyzed with qualitative content analysis. Usage data from eHealth devices were compiled from database entries and analyzed statistically. Parents using the eHealth device expressed reduced worry and stress during the initial period at home through effective and safe communication. Benefits described included keeping track of their child’s progress and having easy access to support whenever needed. This was corroborated by usage data indicating that contact was made throughout the day, and more among families living far away from hospital. The eHealth device potentially replaced phone calls and prevented unnecessary visits. The eHealth technique can aid safe self-treatment within child- and family-centered care in neonatal and pediatric surgery treatment. Future research may consider organization perspectives and health economics.

## 1. Introduction

Complex and costly needs for treatment of children at risk of long-term illness, as in preterm birth and congenital malformations [1,2], are challenging health care systems throughout the world. Aiming to improve child health, interventions are needed to lower morbidity, improve quality of life and decrease costs [3].

EHealth is a broad concept defined by the WHO as “the transfer of health resources and healthcare by electronic means” [4]. Modern eHealth solutions, such as application software attached to devices, are suggested to (1) increase communication between healthcare providers and patients, (2) increase accessibility and patient participation in healthcare [5], and (3) support secure information and data sharing between patients or family, and health-service providers [4]. Pediatric healthcare is rapidly evolving towards outpatient and community care and from health care professionals to parental responsibility [6]. These changes require self-management efficacy and communication support. When done properly, safe individual adapted transitions give the families the best chance to manage their care independently [4]. Potentially, eHealth may replace some of the scheduled or acute in-person consultations, while maintaining equality of access to healthcare.

Globally and in increasing numbers, 15 million babies are born preterm annually [7], and in Sweden just over 5 percent of all children are born prematurely [8]. Providing neonatal homecare (NH) to families has been shown to reduce the length of hospital admission and to unite the family at an earlier stage [9,10]. Still, the transition from hospital to home can be a challenging period for the parents. Therefore, parents and professionals have asked for support through digital communication in addition to home visits and as a replacement for telephone calls [11]. Similarly, in pediatric surgery for congenital gastrointestinal malformations, digital communication including imaging and videos has been reported to be attractive for both patients and professionals [12,13,14,15]. Access to super-specialized medical care implies longer travel distances due to centralization. In addition, times of pandemics may restrict hospital visits and limit opportunities for parental support [16].

Before clinical implementation, eHealth solutions need to be scientifically evaluated regarding safety, but also regarding the extent and quality of the human-computer interaction to ensure that it fits with the contextual needs [17]. The aim of this study was to describe parents’ experiences and usage of an eHealth solution during hospital-to-home transition regarding the child’s condition, parents’ needs, and perceived user-friendliness.

## 2. Materials and Methods

### 2.1. Design

A mixed methods research approach was applied based on both qualitative and quantitative data collection. Using triangulation in a convergent design, qualitative and quantitative data were analyzed separately, and then brought together in the discussion to gain additional insight through comparisons and joint interpretation [18,19]. The study was conducted within the larger context of a controlled experimental clinical trial, based on the Medical Research Council framework for trials of complex interventions [20], evaluating a newly developed eHealth solution provided in addition to usual care after hospital care (ClinicalTrials.gov identifier: NCT04150120).

### 2.2. eHealth Device

The eHealth device was developed in 2016 and refined in 2017–2018, in collaboration with representatives of patient organizations, professionals at the Departments of Neonatology and Pediatric Surgery, and researchers at Lund University [21]. Child-centered care [22] and participatory design (PD) [23] were practiced throughout the development in an iterative process based on continuous collection of feedback from the users involved in the care of the child [21]. Hence, the eHealth device was developed with the perspectives of both professionals and families of patients, and specifically designed for (1) neonatal care, and (2) postoperative follow-up after reconstructions for Hirschsprung disease and anorectal malformations. Corrections of technical issues during development included optimizations for improved battery life and connection reliability. After the development, no additional changes were made during the evaluation for either pediatric surgery or neonatal care.

Delivered as an application for Android-based tablets, the eHealth device could be configured as two different variants, one for each patient group: neonatal and pediatric surgery. A common base of features included video calls and text messaging, enabling chatting with professionals in the hospital. Furthermore, photos could be taken with the tablet’s camera and transferred securely for review by professionals. Exclusive features included registering the weight of prematurely born infants and filling out a report form regarding the well-being and care of infants that had undergone reconstructive surgery. Parents were instructed to report daily on the past 24 h at around 9 a.m. At the other end of the system was the hospital, where professionals could monitor the patients in a web browser from their regular workstation computer, available for communication and response during business hours. The eHealth device operated by communicating wirelessly with a remote, centralized server, where all patient data were stored securely.

The tablets were equipped with SIM cards that provided a data plan to ensure uniform network access and were locked down using third-party software for security reasons. No functions other than the eHealth application were available, e.g., no web browser or games. Images were automatically deleted from the device after being sent. The eHealth devices were intended to be “always-on” for interruption-free operation and timely notifications and communication.

### 2.3. Participants and Setting

The study was conducted between August 2019 and June 2020 at a neonatal department and a national center of pediatric surgery, at a university hospital in the south of Sweden, covering a catchment area of 2 and 5 million residents, respectively, with travel distances of up to 1000 km. The study coincided with the coronavirus pandemic, with national restrictions beginning in Sweden in March 2020 [24].

Participants were included based on systematic probability sampling during a fixed period within the main study. Consecutively included families in each clinical area were invited to participate based on predefined inclusion criteria, including legal guardians (parents) who were able to read and write Swedish with children below four years of age, who were planned for discharge after advanced hospital treatment for prematurity or surgery for congenital colorectal malformations. According to a structured oral information procedure and written information sheet, parents were instructed about the features of the eHealth device, how to use it, and about how the communication with the professionals would be organized.

Table 1 provides a description of 21 children (including three pairs of twins) from 18 out of 20 consecutively recruited families who received eHealth devices and where at least one parent was interviewed about their experiences of having access to the device at home after hospital discharge. All participating parents were married or cohabitant. Eleven out of 16 interviewed parents who reported the educational level had studied at university level. Ongoing treatment and needs of home-based care after hospital discharge included managing nutritional needs, tube feeding, operational wounds, stomas, bowel management, medications, and prophylactic treatment for perianal skin erosions. Recruited families were invited to use the application on the eHealth device for communication and support from professionals after leaving the hospital, in addition to traditional contact channels (telephone and physical meetings) and routine in-person follow-up controls. Time at home with the eHealth device varied from two to five weeks.

### 2.4. Data Collection

Data collection included compilation of database entries from the usage of eHealth devices, along with qualitative interviews.

Statistics on usage of the eHealth device were compiled from the database entries that had been created each time the parents had used the device. Data were collected from 18 out of 21 devices, nine from neonatal and nine from pediatric surgery. Three eHealth devices were excluded since all three families with twins used only one of their two assigned devices. Usage data included the number of individual chat messages sent between the families and the professionals at the hospital and the number of images taken by the families and sent for review by professionals at the hospital. Furthermore, features of each group (prematurity or surgery) were analyzed, including daily reports and the number of weight measurements entered by the families and sent for review by professionals at the hospital.

Two to eight weeks after the parents had returned the eHealth device, they were contacted for an individual semi-structured interview. Contact was made by email followed up by telephone calls. Interviews were performed by the first author (RL) who had never been involved in the child’s care and was unknown to the parents. Each interview was initiated with an open question about the overall emotional experience of arriving home after leaving the hospital. An interview guide, tested with two pilot interviews, was used, containing examples of open questions relating to emotional and practical perspectives of caring for a child after hospital discharge, including needs and wants while having access to the eHealth device for healthcare communication. The pilots, which were included in the analysis, resulted in some minor changes in the introductory questions of the interview guide, aiming to further enhance contact and gain insight into the families’ background situation prior to discharge. Parents were invited to share experiences, positive and negative, during the period they had access to the device, to provide examples of how they had used it and to say what they thought it had meant to them in different situations. Overall, interviews were performed with the intent to follow the participant with relevant follow-up questions to gain depth through exploration of perceptions and details of the experiences shared. Lastly, parents were asked to share thoughts on adaptations and improvements of an eHealth device for others in similar situations. Interviews lasted a median 13 min (7–39 min) and were performed over the telephone.

### 2.5. Data Analysis

Interviews were transcribed and analyzed with qualitative content analysis on a latent level, in a text-driven inductive approach [25]. The authors, three of whom had extensive previous experience in qualitative research, read and reread the interviews several times to get an idea of the overall meaning and identified meaning units, condensed, and coded. The differences and similarities among various codes were compared during discussions and reflections between the authors. The approach to interpretation was a movement from individual details to the general. Codes were grouped into subthemes and themes until joint agreement was reached. Subthemes are exemplified below with quotations marked with a number indicating participant, where PS are parents of children treated at the Department of Pediatric Surgery and N are parents of children treated at the Department of Neonatology. Qualitative data were managed using the software Open Code [26] and reporting followed the standards for qualitative research (SRQR) [27]. Data from eHealth devices were analyzed with descriptive statistics.

### 2.6. Ethical Considerations

This study was approved by the Swedish Ethical Review Authority (no. 2019-0341). Personal data were handled confidentially. The intervention was provided as a complement to usual care. Thus, no usual care was excluded. Participating families were able to drop out without explanation and without any effects on usual care. None of the authors performing data collection and analysis were involved in the care of the children.

## 3. Results

### 3.1. Parent’s Usage of eHealth Devices

Analysis of database entries for statistics on usage of the eHealth devices (Table 2) showed the demand for support and communication. The eHealth device was used for median 2.5 (2–5) weeks. Parents in the pediatric surgery group compared to parents in the neonatal group sent more images/device (median 6 (0–15) versus 0 (0–2)) and chat messages/device (21 (5–36) versus 2 (0–11)). With respect to the exclusive features of the devices (daily reports and weight registrations) these were also used more by parents in the pediatric surgery group (median 13 (0–28)), although weight registrations were the most frequently used function in the neonatal group (5 (0–16)). Data before and during the pandemic restrictions in Sweden indicated no change in usage regarding the number of contacts (chat messages, images, weight registrations, and daily reports) per user per month. (Data not shown.)

Table 3 shows the total average usage of the eHealth device and the distribution of use over 24 h (morning, noon, afternoon, evening, and night) and weekdays. Most of the communication through the device took place during business hours on Mondays to Fridays. In the pediatric surgery group, parents most frequently sent chats and images during business hours and on weekdays, while their daily reports were distributed equally on all weekdays. In the neonatal group, the usage was concentrated on weekdays. Weight reports were generally sent in the morning or early afternoon, while chat messages were sent during business hours. The timing of daily reports was spread over the day with around half of all reports (56/120) within a timespan of ±1 h of the target time 9 o’clock.

### 3.2. Parents’ Experiences of eHealth Devices (Interviews)

Parents experienced the hospital-home transfer differently. Overall, they felt relief over leaving the hospital environment to come home. Some parents reported feeling secure and certain about how to care for their child/ren, while others said they felt burdened by responsibility and insecurity. Regarding communication through the eHealth device, parents’ experiences of the hospital-to-home transition revealed three themes with sub-themes (Table 4).

#### 3.2.1. Having Easy Access

Parents expressed appreciation for receiving a direct link to hospital support through modern and informal communication from the living room sofa. The device was viewed as a natural component of modern healthcare which replaced and complemented other contact channels with healthcare.

##### Accessing Familiar and Up-to-Date Ways of Communication

Parents said that they had used the device typically for non-acute questions concerning minor worries or practicalities, such as asking for medical supplies, booking meetings, or asking about dosage. Acute issues or complex concerns required instant feedback through visits or phone calls. The chat was referred to as an ongoing dialogue, where a question regarding, for example, signs of infection, time between feedings or poo color did not have to be important or urgent to be legitimate. This was contrasted to phone calls, which needed to be well-prepared with relevant questions to be “worthwhile.” Parents stated that they found the device supported them in caring for their child more effectively and handling their own worry. Using images instead of describing in words what something looked like was mentioned as especially valuable.


*“Many of the questions I asked there [in the chat] I wouldn’t have called about. Because, like, if you are going to call them it has to be about something real.”*
(N24, mother)

The device was a much appreciated alternative to some in-person visits with troublesome and risky journeys or home visits requiring house cleaning. By using video counselling or image transfers, combined with phone calls, parents could get a professional opinion regarding the potential need for a follow-up visit, which was also appreciated in relation the current corona pandemic.


*“Instead of going into the hospital at this time with a lot of corona, you could have a video conference. And then we sat here in the sofa talking to the surgeon and the nurses.”*
(PS19, father)

##### Feeling Less Stressed in Communication

Parents described their contact with healthcare as less stressful, when being able to consider their own communication preferences, while ensuring respect for the nurses’ working situation. Being part of the “tablet generation” was equivalent to being used to electronic devices as well as being uncomfortable with talking over phone compared to texting. Taking care of a small baby with special care needs, while waiting in telephone queues or having to remain available for returning calls, could be quite stressful.


*“You don’t have to be stuck on the phone like that. Instead, you send it off. And later you receive an answer.”*
(N8, mother)

The chat provided time to phrase questions, while allowing for interruptions, such as soothing, changing a diaper or breast-feeding. Parents stated that they were less hesitant about reaching out knowing that their question could be sent off at any time, picked up and resolved later without interrupting important work performed by the nurses.


*“You are able to spend ten minutes without feeling like you are taking up someone else’s time while phrasing a correct message. […] It felt very good to be able to have a form of communication where you know you’re not disturbing anyone.”*
(PS16, father)

Waiting for an answer could be a struggle due to the uncertainty of not knowing when it would arrive. Therefore, parents suggested that time limits for response should be communicated clearly. Parents judged that the waiting also meant that the answer was more likely to be correct and clear once it arrived. It made a difference to them to be able to see when their message had been read.


*“The fact that they had seen the message. That gives great security. You know that your question is being handled.”*
(PS24, father)

#### 3.2.2. Relying on Safe Technology

Parents highlighted the value and need for effective and safe eHealth communication in terms of being functional, practical, and secure.

##### Reflecting on User-Friendliness and Technical Hurdles

The perceived effectiveness of the device as a communication tool was affected by individual experiences of technical challenges, the introduction to the device, the interactive design, and user-friendliness. The device was described as easy to use and intuitive, which was important in their situation of caring for a child.


*“When you have little ones, everything should preferably go as quickly as possible. And all the buttons were pretty much on the same screen.”*
(N1, mother)

Downsides of the eHealth device and its use were also described. Some parents said that they never got started with the device. They shared reflections on how nurses did not have time or technical skills to solve issues—which was emphasized as understandable, since this is outside their area of expertise. Parents also expressed their understanding and expectations of programs and electronical devices causing technical obstacles, needing restarts, or even getting stuck.


*“You know that technology is what it is. It doesn’t always work.”*
(PS13, mother)

The poor battery was brought up as a problem, causing parents to miss messages due to the device being stuck to the plug in another room, but this was also a minor issue in relation to the value of having the device at all. Still, lost connection and failed login affected the perceived overall usability and level of integration of the device in their everyday life. The technical quality of video meetings ranged broadly from working smoothly to giving rise to repeated issues or not working at all.


*“We knew she was going to call. And then we just answered, and it worked like a charm. And it was great picture and sound.”*
(N27, mother)


*“We were able to answer but then nothing happened after that. And we really tried many, many times. But no, we were simply never able to talk. At all.”*
(N24, mother)

Parents expressed a great belief in the potential for further development of the eHealth concept, especially through video communication. They also identified a need to establish routines for users and health care professionals to make sure that the technique could be more effectively utilized.


*“It felt a bit unnatural, I guess. And I think that everyone felt a bit unaccustomed to the technique.”*
(PS16, father)

Parents suggested general improvements of the device, such as a protective cover and possibilities to link to their child’s medical journal within the device. They specifically wanted the device to contain more features of a modern smartphone, including improved touch sensitivity, possibility to change layout format, notification sounds and improved image quality.

##### Emphasizing the Importance of Cybersecurity

Encrypted data in relation to sending intimate images of sensitive parts of their babies was highlighted as important.


*“Why, we photographed [child’s] bottom. To take a shot of such a sensitive area of your child and send it to someone else. It doesn’t feel great to know that it may end up in the hands of just anyone. However, this was for the nurses who would look at it and then erase it.”*
(PS13, mother)

The fact that images disappeared after being sent, was experienced differently: from those who thought it was worth it for safety reasons, to those who suggested that the images should remain in the device until returned and emptied. The image deletion function for safety was reported to have made consultation with healthcare staff difficult.


*“We may have forgotten exactly what the image looks like. And then they have it on the other side, but we can’t see it anymore. (…) Perhaps they’re talking about some shadow in the image [that we no longer can see].”*
(PS15, mother)

#### 3.2.3. Sensing Support

Parents described the device in terms of enabling healthcare to remain present at home. Possibilities to use it to fulfill ad hoc needs, as well as be involved in a structured clinical follow-up, created a sense of trust.

##### Experiencing Shared Responsibility

Staying connected and in continuous communication with care professionals who were known to the child since the in-hospital stay, parents described experiencing a sense of security during the hospital-to-home transition thanks to the device. This was something “like taking the red button home with you” (BKI5), so that going home was perceived as easier and less worrying. Parents said that they had sometimes felt that the device was more for their own sake than for the health of the child, increasing their emotional security.


*“We always had the possibility to ask ’Is this normal?’ And then it felt so good to hear ’Yes, it’s normal’. And then we could continue with our day.”*
(PS6, father)

The device was described as making parents feel prioritized, emphasizing the value of receiving individually adapted responses or counterquestions. Parents also shared opposite experiences of physicians keeping video meetings short because they had to attend to other tasks. Still, the palette of chat, images, video calls, and standardized reporting supported two-way communication, where responsibility was shared while gradually transferred from healthcare to parents at their own individual pace.


*“We had the responsibility. But so did they. Yes, it felt like we were sharing responsibility, you might say.”*
(N24, mother)


*“In a way we took the hospital home with us. It was really nice. So, the contact did not end just because we left. (…) It felt very safe to have such close contact once we were home.”*
(PS15, mother)

Parents were generally content with the support they received and the length of time they had access to the device, saying that it fit with their needs. Parents also expressed that returning the device was like cutting the cord, when no longer being able to access the medical support in the same way.

##### Keeping Track and Following Progress

Parents experienced how the device contributed structure and formal reporting, providing reassurance in “black on white” with early detection of signs of the child’s health state, by keeping track of weight, feeding, medications, and diapers.


*“In a phase where a child has growth inhibition the focus is on them growing properly. It is very reassuring for us as parents—it becomes very visible to us—how the development was with the weight curves and such.”*
(N6, father)

Parents expressed a huge interest in reading the reported data, suggesting that data could be continuously summarized, displayed, and followed, for example, data on the children’s mood and diapers. The device kept a permanent record, available to be read again or, for one parent, to see what the other parent had reported or communicated on. Questions in the daily reports regarding, for example, medication were not always perceived as relevant to their specific case or adapted for persons not working within healthcare, e.g., names of medications. Being monitored was put forward as a prerequisite for being able to go home and not stay at the hospital longer than necessary, which was highly valued. The formality of the daily reports with feedback from healthcare created experiences of being followed closely through continuous reporting.


*“You really felt that they went in and really checked every day, what we had answered and, like, how his general condition was.”*
(PS2, father)

## 4. Discussion

This study demonstrated that parents of children with prematurity or in need of pediatric surgery overall experienced the access to and use of eHealth device positively in the transfer period between hospital and home. According to the parents, the eHealth device supported their self-treatment by providing valuable options for secure and up-to date communication with health care professionals, in addition to traditional channels of communication. It also provided parents with valuable options to raise questions and handle concerns through a preferred medium, such as spoken words, video, text, or images, with consideration of the situation at hand. Results of the interviews were supported by statistics on usage in relation to the child’s condition and distance from home to hospital, in that usage was highest among families from pediatric surgery, who were managing operational wounds as well as living further away from hospital. This study adds to the limited research on eHealth within pediatric surgery and neonatal care, and especially regarding the ongoing centralization of specialist care and during pandemics. Results of qualitative interviews with parents and data on eHealth usage may be linked to several aspects of eHealth in relation to communication, technology, and organization.

Parents felt that the possibility to use the eHealth device for communication and information exchange through images and data reports enabled professionals to keep track of the children’s progress, without being dependent on the parents’ ability to interpret or explain. Hence, the device was less affected by potential differences in knowledge and views between parents and healthcare professionals when sending, receiving, and responding to messages [28]. The voluntariness provided a basis for using eHealth for support whenever needed, avoiding any unwanted control by health care. Results from interviews as well as statistics on the numbers of messages and images over 24 h, corroborate that the eHealth device enabled flexibility and a facilitated equal and effective communication. This study thereby adds to the knowledge of ethical implications of human-computer interaction in relation to systems in use and design [29].

A potential barrier to the successful introduction of eHealth is the establishment of the necessary technical infrastructure for electronic transmission of health care data [30]. Providing each family with a physical device with its own SIM card (instead of, for example, an app for mobile phones) meant uniform network access for all participating families, not dependent on existing devices at home or a good internet connection. Some of the technical challenges, such as poor battery and change of SIM card, were present in the initial phase of the study and were then resolved. Yet, there were still families with no or low eHealth usage, which indicates the importance of identifying and addressing other potential barriers to eHealth to minimize the risk of unequal access [31]. Ethical perspectives of eHealth in relation to data confidentiality and security, which are important to guarantee safe use [32], were mentioned as crucial by the parents, especially in relation to secure transmission of sensitive images of their children.

Establishing the necessary administrative and operational infrastructure for new eHealth initiatives is a potential hurdle to achieve optimal use of eHealth [30,33]. The results of this study suggest potentials for further improving the organizational change process, for example, in defining when a nurse should provide technical support. Data on eHealth usage showed that contact was made by the parents less on weekends but during weekdays almost equal throughout the day’s hours. This indicated the benefit of not having to adapt to, say, two-hour time slots for telephone hours. The benefit of access to support whenever needed or convenient was also highlighted by parents in the interviews. Total usage over part of the day and over the weekdays is important for planning the need to have healthcare professionals available to manage communication before general implementation. Geographical spread is a known potential barrier to equal access to visiting specialists which may be addressed with the use of eHealth [34,35]. Since our study included families with up to 365 km distance from home to hospital, there was an obvious benefit of reducing or replacing physical healthcare visits, which could be a gain, especially during pandemic times [36]. Usage data confirmed the benefit of eHealth in relation to distance from home to hospital, which was also mentioned in interviews, as usage was higher among families living further away. However, usage data indicated no change in eHealth usage in relation to the corona pandemic.

Since cost is the most frequently mentioned factor contributing to failure of eHealth interventions [37], it is important to evaluate the cost-effectiveness of new eHealth initiatives to secure proper investment. Such an analysis needs to include costs to society, including effects on overall healthcare consumption in relation to all benefits. In this study, notions of the device as “taking the hospital home” suggest that access to the device contributed to a family- and child-centered approach to care [2]. The eHealth device enabled contact with healthcare, while parents were able to remain sensitive to their child’s emotional and physical needs, allowing for interruptions to respond to needs of comfort, food, or care. Parents felt strengthened in their self-efficacy in caring for their child, in partnership with healthcare professionals within the context of family and home—the child’s right environment.

### Limitations

This study has several limitations. Usage data did not include video calls since these were not permanently saved on the devices. In addition, the intervention was not randomized, and thus, we were not able to evaluate the end effect. Gender-dependent usage could not be analyzed since the devices and login were provided per family.

Participants represented mothers and fathers with experiences of either having a prematurely born infant or having gone through surgeries. Both parents were interviewed in some families but not in all, which might have caused a representation bias. The study did not include families who did not speak Swedish, and therefore, a cultural skewedness cannot be excluded. Although the intervention may potentially influence all family members, the analysis did not include considerations with regard to siblings which, in retrospect, could have added substantial information. In addition, since it was difficult for some parents to concentrate on the interview as the child or siblings were nearby, needing assistance, some interviews become short. Two months had passed since parents had used the eHealth device when they were interviewed, which might have caused difficulties in remembering details or emotions.

## 5. Conclusions

In its most successful form, the provision of the eHealth device for support appears to have struck a balance between healthcare remaining present and supportive, yet not intrusive or overwhelming, where parents could feel safe caring for their children while exercising and developing their parental role at home. This is supported by usage data demonstrating flexibility and availability, as contacts were made throughout the day. Parents’ experiences imply that eHealth may be effective by leading to increased self-efficacy and quality of life. Future research may take organizational and health-economic perspectives into account.

## Figures and Tables

**Table 1 ijerph-18-06480-t001:** Background characteristics of 21 children and 25 parents from 18 families that received eHealth devices and where at least one parent was interviewed.

Children Treated at the Department of Pediatric Surgery (*n* = 9)	
Surgical procedures	
Transendorectal pull through for Hirschsprung disease	3
Posterior sagittal anorectal plasty for anorectal malformations	3
Appendectomy, laparotomy	1
Anorectal reconstruction	1
Reconstruction for Esophageal Atresias	1
Age at hospital discharge in weeks, median (range)	5 (3–162)
Weight at hospital discharge in grams, median (range)	4014 (3100–14000)
**Children treated at the Department of Neonatology** (***n* = 12 ^1^**)	
Gestational age at discharge in weeks, median (range), *n* = *6*	35 (34–36)
Days since birth at discharge, median (range), *n* = 6	7 (6–7)
Weight at hospital discharge in grams, median (range), *n* = 10	2104 (1770–2615)
**Distance from home to hospital in km, median** (**range**)	
All (18 families)	22.5 (8–365)
Discharged from Department of Neonatology (9 families)	18.5 (8–60)
Discharged from Department of Pediatric Surgery (9 families)	280.5 (20–365)
**Age and gender of interviewed parents** (***n* = 25 ^2^**)	
Female, *n* (percent)	14 (56)
Male, *n* (percent)	11 (44)
Age, median (range)	31 (28–42)

^1^ Including three pairs of twins. ^2^ Representing in total 18 families. In seven families both parents were interviewed.

**Table 2 ijerph-18-06480-t002:** Per family-usage of eHealth device, including data obtained from 18 eHealth devices provided to 18 families, whereof 9 from pediatric surgery and 9 from neonatal care. The eHealth device was used median 2.5 (2–5) weeks per family.

	Average	Min	Median	Max
**Pediatric surgery**				
Number of messages sent	19.9	5.0	21.0	36.0
Number of images taken	6.6	0.0	6.0	15.0
Number of daily reports	13.3	0.0	13.0	28.0
**Neonatal**				
Number of messages sent	3.1	0.0	2.0	11.0
Number of images taken	0.3	0.0	0.0	2.0
Number of weights registered	6.7	0.0	5.0	16.0

Definitions: Messages: the number of individual chat message sent by the families from the eHealth devices to professionals at the hospital. Images: the number of photos taken by the families and sent for review by professionals at the hospital. Daily reports: the number of report forms that were partially or completely filled in on the eHealth device and sent for review by professionals at the hospital. Weight registered: the number of weight measurements that were entered by the families on eHealth devices tablets and sent for review by professionals at the hospital.

**Table 3 ijerph-18-06480-t003:** The distribution of eHealth device usage over 24 h and weekdays (Monday is denoted 1), including data obtained from 18 eHealth devices provided to 18 families, whereof 9 from pediatric surgery and 9 from neonatal care. The eHealth devices were used during median 2.5 (2–5) weeks.

	Function	Sum by Part of Day (24-h Clock)					Sum by Weekday						
		04–08	08–12	12–18	18–23	23–04	1	2	3	4	5	6	7
**Pediatric surgery**	Number of messages sent by patients	4	69	96	8	2	45	32	35	33	29	3	2
	Number of images taken by patients	13	25	13	8	0	12	11	8	11	8	5	4
	Number of daily reports by patients	8	63	37	10	2	24	20	15	18	12	17	14
**Neonatal**	Num chat messages sent by patients	1	18	8	1	0	5	9	4	6	3	0	1
	Number of images taken by patients	0	2	1	0	0	0	0	0	0	3	0	0
	Number of weights registered	13	34	9	4	0	9	12	6	6	17	4	6

Usage of functions over an average day and an average week. Definitions: Messages: the number of individual chat message sent by the families from the eHealth devices to professionals at the hospital. Images: the number of images taken by the families that were sent for review by professionals at the hospital. Daily reports: the number of report forms that were partially or completely filled in on the eHealth devices and sent for review by professionals at the hospital. Weights registered: the number of weight measurements that were entered by the families on eHealth devices tablets and sent for review by professionals at the hospital.

**Table 4 ijerph-18-06480-t004:** Themes and sub-themes of parents’ experiences of communicating with healthcare professionals through an eHealth device after hospital discharge.

Having Easy Access	Relying on Safe Technology	Sensing Support
Accessing familiar and up-to-date means of communication	Reflecting on user-friendliness and technical hurdles	Experiencing shared responsibility
Feeling less stressed in communication	Emphasizing the importance of cybersecurity	Keeping track and following progress

## Data Availability

The data presented in this study on usage of the eHealth device are available within the article. Interviews supporting the findings of this study are not publicly available due to privacy.

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
