# Peer review of "“In a Way We Took the Hospital Home”—A Descriptive Mixed-Methods Study of Parents’ Usage and Experiences of eHealth for Self-Management after Hospital Discharge Due to Pediatric Surgery or Preterm Birth"

_ijerph, 2021, doi:10.3390/ijerph18126480_

Round 1

Reviewer 1 Report

What a wonderful report and study!  This is well written and very readable. The protocols are  described consistently and completely.  Your mixed methods are used as they should be. Content analysis as qualitative method was used as intended. The qualitative and quantitative results complement and expand on each other. 

Thank you,  I hope this intervention spreads and scales rapidly. 

Author Response

Dear reviewer,

We are happy to see that the manuscript was well received. It is our pleasure to hereby submit our revised manuscript ijerph-1218861. The revised manuscript is submitted with highlighted changes which we think further improved the manuscript. Thank you for constructive feedback! 

Rose-Marie Lindkvist

Reviewer 2 Report

Dear Authors

Thank you for this interesting, well-written, and valuable contribution to the use of eHealth devices in children's health care. I do have some questions. The introduction is wordy and could benefit from a more stringent approach towards the aim of the study. Let me suggest: What is the problem, what will the benefits be if the problem is solved and how can we solve the problem. 

Design, row 71, mixed-method design, could you elaborate on what mixed-method design and the strengths in relation to your study? 

Row 78, did anything change from development 2016 in 2017-18? Row 81. How were child-centered care and participatory design applied?

In the participant setting: Could you develop on the families, which families accepted to be a part of the study? where there a difference in education or were they a homogenous group? 

Row 154 pilot interviews, was anything changed? why? were they included in the study? Row 161, what does depth mean in this study? 

Under the ethical considerations, I would like you to develop how the use of the device might affect the family in a broader sense, for example, integrity, equality, feeling of unwanted control, support, or reflections upon what it might mean to their developing role as parents.

Discussion

Were there any differences between mums and dads? are there any thoughts about who will manage or who can not benefit from this device? Are there any considerations relating to older or younger siblings?

row 443 what does rich in content mean here? 

Thank you for letting me read this interesting paper.

Author Response

Dear reviewer,

It is our pleasure to hereby submit our revised manuscript ijerph-1218861. Your comments and suggestions were taken into careful consideration and has resulted in changes that we think improved the manuscript. The revised manuscript is submitted with highlighted changes. The attached table summarizes our considerations of your comments and explains revisions that were made. We are very thankful for constructive feedback.

Rose-Marie Lindkvist

Reviewer 3 Report

Overall the authors should be congratulated for attempting to draw together a complex picture of parental experiences with an eHealth device for neonates. This is an important and useful study which should be available in the literature, however changes to the manuscript are required for it to be acceptable for publication.

The language can be streamlined and improved for readability--some sentences are awkwardly phrased and/or overly long and confusing.

A major concern that must be addressed by the authors is that the study design is not appropriately defined in the Methods section under "Design" and is conflated with the overall intervention study of which this study is a part. In other words, the study is misleadingly described as "performed as a controlled experimental study" when in fact it is a descriptive (observational/cross sectional mixed methods study) conducted within the larger context of an intervention study or trial. This must be corrected and is a very important distinction. It's also recommended to change the title to include the study design, e.g. "In a way we took the hospital home with us": A mixed method observational/descriptive/cross sectional study of parental experiences of eHealth..."

The quantitative data are not very helpful nor as useful in terms of understanding the parental experience. Perhaps explaining why they are included or what their relevance are will help with this (is there some literature from CHI or human-computer interaction studies that can situate the data?) Tables 2 and 3 are gravely in need of some context in order to be made meaningful to the reader. These counts of interaction with the device mean what? What are we meant to interpret? Some parents are simply using the device a lot and others are not. What is the importance of time of day and other metrics presented--this must be explained. Can denominators be added in the tables to help make sense of the data?

For the qualitative portion, the authors need to add information from the COREQ or SRQR qualitative checklists and to cite one of those to fill out the methods and improve trustworthiness. A guiding theoretical approach needs to be mentioned that drives the approach to "content analysis" in qualitative research. If sampling was purposive this needs to be clearly stated. Information should be added to clarify researchers previous qual experience, approach to interpretation in content analysis, etc..  Please see COREQ/SRQR items that should be added to the manuscript.

Discussion section needs extensive revision before publication. The manuscript needs a Limitations section, and the paragraph titled "Methodological considerations" needs to be removed or revised greatly. It appears to be highlighting several perceived strengths of the paper that are not accurate. The sentence "The combination of qualitative and quantitative methods strengthens the validity of the results by providing broad as well as deep insights based on joint interpretation of objective and subjective data" is not supported and should be removed (the abstract and somewhat meaningless quantitative data that is presented without context is not integrated with the qualitative data nor does it provide broad/deep insights or joint interpretation). The sentence "Credibility of the qualitative data collection and analysis was ensured by a thorough systematic analysis."  should also be removed The sentence "Four authors read the interviews and the analysis and discussed the subthemes and themes until consensus was reached" is not relevant nor meaningful here because there was no guiding theoretical qualitative paradigm indicating that consensus approach would be the goal of qualitative analysis. Indeed many approaches to qualitative research eschew consensus--and if consensus was the goal it was not measured by an objective coefficient. If this was the approach it needed to have been mentioned in the Methods section alongside relevant references to using consensus. "All parents were consecutively invited." should also be removed, or revised, to indicate how this was a strength or added to the validity. It seems to contradict earlier text given that the Methods section described inclusion and exclusion criteria. Overall the Methods and Discussion sections are not coherent.

Finally, there is a huge lack of citation and reference to any of the vast human computer interaction literature that should be included in the Introduction and Discussion section. This also needs to be addressed in revisions.

Author Response

(The authors gave the same response as above.)

Round 2

Reviewer 3 Report

The revisions have been done to a high standard and the manuscript will add value to the literature. The authors have done well in their presentation of the material.